# Catalytic activity imperative for nanoparticle dose enhancement in photon and proton therapy

Lukas R. H. Gerken[1,2], Alexander Gogos[1,2], Fabian H. L. Starsich[1,2], Helena David[1], Maren E. Gerdes[1], Hans Schiefer[3], Serena Psoroulas [4], David Meer[4], Ludwig Plasswilm[3,5], Damien C. Weber[4,5,6] & Inge K. Herrmann [1,2✉]

Nanoparticle-based radioenhancement is a promising strategy for extending the therapeutic ratio of radiotherapy. While (pre)clinical results are encouraging, sound mechanistic understanding of nanoparticle radioenhancement, especially the effects of nanomaterial selection and irradiation conditions, has yet to be achieved. Here, we investigate the radio-enhancement mechanisms of selected metal oxide nanomaterials (including $SiO_2$, $TiO_2$, $WO_3$ and $HfO_2$), TiN and Au nanoparticles for radiotherapy utilizing photons (150 kVp and 6 MV) and 100 MeV protons. While Au nanoparticles show outstanding radioenhancement properties in kV irradiation settings, where the photoelectric effect is dominant, these properties are attenuated to baseline levels for clinically more relevant irradiation with MV photons and protons. In contrast, $HfO_2$ nanoparticles retain some of their radioenhancement properties in MV photon and proton therapies. Interestingly, $TiO_2$ nanoparticles, which have a comparatively low effective atomic number, show significant radioenhancement efficacies in all three irradiation settings, which can be attributed to the strong radiocatalytic activity of $TiO_2$, leading to the formation of hydroxyl radicals, and nuclear interactions with protons. Taken together, our data enable the extraction of general design criteria for nanoparticle radio-enhancers for different treatment modalities, paving the way to performance-optimized nanotherapeutics for precision radiotherapy.

[1] Nanoparticle Systems Engineering Laboratory, Institute of Energy and Process Engineering (IEPE), Department of Mechanical and Process Engineering (D-MAVT), ETH Zurich, Sonneggstrasse 3, 8092 Zurich, Switzerland. [2] Particles Biology Interactions Laboratory, Department of Materials Meet Life, Swiss Federal Laboratories for Materials Science and Technology (Empa), Lerchenfeldstrasse 5, 9014 St. Gallen, Switzerland. [3] Department of Radiation Oncology, Cantonal Hospital St. Gallen (KSSG), Rorschacherstrasse 95, CH-9007 St. Gallen, Switzerland. [4] Center for Proton Therapy, Paul Scherrer Institute (PSI), Forschungsstrasse 111, 5232 Villigen PSI, Switzerland. [5] Department of Radiation Oncology, University Hospital Bern (Inselspital), 3010 Bern, Switzerland. [6] Department of Radiation Oncology, University Hospital Zürich, 8091 Zürich, Switzerland. ✉email: ingeh@ethz.ch

Radiation therapy is an integral part of cancer treatment and is applied with at least 50% of all cancer patients[1,2]. This treatment modality has low tissue specificity, and despite considerable advances in dose delivery, healthy tissues in vicinity of the target volume usually receive undesirable radiation doses, potentially leading to significant side effects[3]. Generally, containment of the late toxicity to healthy tissues determines the maximum dose that can be delivered to the tumor during radiotherapy. To overcome the aforementioned limitations and increase the therapeutic ratio, nanoparticles offer a promising route to targeted radiotherapy by acting as radioenhancers[4]. Nanoparticles deposited in the tumor tissue selectively increase the radiation absorption cross-section relative to that of healthy tissue surroundings[5]. The effect of ionizing radiation on biological structures is governed by physical, chemical, and biological phenomena[6,7]. The exact contributions of nanoparticles, and their material composition in particular, during these stages and within a cellular environment during irradiation is yet to be understood. The current mechanistic understanding is especially hampered by the lack of fundamental and comparative studies[4,8], which precludes rational nanoparticle radioenhancer design.

Considering physical dose enhancement only, high-$Z$ nanoparticles are a natural choice since their photoelectric absorption cross-section, scaling approximately with $Z^4$, is significantly higher than those of soft tissue or water[9]. However, the photoelectric tissue contrast is also strongly dependent on the energy of the incoming photons ($\sim E^{-3}$). Hence, and in contrast to kV X-rays, only limited dose enhancement would be expected at higher energies (MV X-rays)[9,10]. In fact, at energies exceeding 500 keV, physical interactions are dominated by Compton scattering events with cross-sections linearly proportional to $Z$[11]. Therefore, it has been suggested that chemical and biological effects play pivotal contributing roles in the nanoparticle dose enhancement found in vitro and in vivo with MV photons[7,9,12]. There is increasing experimental and clinical evidence supporting nanoparticle-based radiotherapy enhancement in terms of safety and efficacy for both kV and MV photons[8,13–16]. Most notably, $HfO_2$ nanoparticles marketed by Nanobiotix as NBTXR3/Hensify® have recently gained approval for the European market[16]. These $HfO_2$ nanoparticles obtained European CE Mark approval in April 2019 for the treatment of locally advanced soft tissue sarcoma via intratumoral injection with photon radiotherapy and are being investigated for treatments of other cancers[17].

While surprisingly little is known about nanoparticle radioenhancement mechanisms with clinical MV photon beams, nanoparticle dose enhancement using protons has been even less investigated. Protons can also be used as alternatives to photons in treating cancers and exhibit better dose conformation. As positively charged subatomic particles, protons interact differently with matter, leading to a distinctly different dose–depth profile compared to those of noncharged photons[18,19]. While the dose deposition of photons in depth is continuous and goes beyond the tumor, resulting in an "exit dose", protons lose the majority of their energy in the range of the Bragg peak, after which they are stopped completely[19]. In the Bragg peak region, protons are slowed down, leading to an increase in the interaction probability with orbital electrons and the number of ionization events in the tissue. Finally, the proton is absorbed in a charge-changing process[18]. The position of the Bragg peak can be tailored to the location and size of the tumor volume, to minimize the dosage delivered to the surrounding tissue[19]. With regard to nanoparticle proton therapy enhancement, several Monte Carlo studies have investigated the influence of proton energy[20], nanoparticle size and coating[21], or nanoparticle clustering[22] on the dose-enhancement mediated by Au nanoparticles. Additionally, in vitro dose-enhancement studies for Au nanoparticles used with proton irradiation have been performed, indicating dose enhancements in the range of 0–44%[23–27]. Moreover, Fe-based or Au nanoparticles in combination with proton irradiation were successfully used to achieve complete tumor remission in vivo[28].

While the above studies show generally promising results for dose enhancement by nanoparticles, they also suggest a strong dependence on nanomaterial composition and the type of irradiation. This has also been indicated by the findings of Smith et al. (2015), who compared the dose enhancement by Au nanoparticles during X-ray (kV and MV) and proton irradiation in alanine electron paramagnetic resonance (EPR) dosimeters, showing an approximately 60% enhancement for kV, 10% for MV and 5% for proton irradiation[29]. Despite individual successes, including the clinical use of $HfO_2$ nanoparticles, we lack a mechanistic understanding and thus fail to select nanoparticle materials for optimal dose enhancement under various irradiation settings (photons, protons) in an evidence-based manner.

Here, we present a conceptual investigation which disentangles key contributors to nanoparticle dose enhancement as a function of nanomaterial composition and beam characteristics. We synthesized a portfolio of nanoparticles made from different core materials (including TiN, $TiO_2$, $WO_3$, and $HfO_2$, as well as Au and $SiO_2$ as commercially available reference materials) and assessed their physical, chemical, and biological dose-enhancement capabilities in consistent settings for which observed differences can be directly attributed to nanomaterial properties as the only variable in the system. We demonstrate that Au nanoparticles show excellent dose enhancement in kV irradiation settings, that vanish almost completely in MV photon and proton therapy settings. In contrast, $HfO_2$ nanoparticles retain some dose-enhancement capability in MV beam therapy settings, even though greatly reduced compared to kV irradiations. Most interestingly, catalytically active $TiO_2$ nanoparticles retain their dose-enhancement effects in all three irradiation settings.

## Results and discussion

**Synthesis and physicochemical characterization of nanoparticle radioenhancer.** Figure 1 shows transmission electron microscopy images (TEM) of the flame spray pyrolysis (FSP) synthesized ($TiO_2$, TiN, $WO_3$, $HfO_2$) and commercially available ($SiO_2$, Au) nanoparticles used for radioenhancement investigations. Primary particles had spherical morphologies, with the exception of TiN and $WO_3$, which presented slightly elliptical shapes. The mean primary particle diameters based on TEM images ($d_{TEM}$) were ~5 nm for $TiO_2$ and $HfO_2$, 10–15 nm for $WO_3$ and TiN and 50 nm for Au nanoparticles (Table 1).

Crystal sizes obtained from XRD measurements ($d_{XRD}$) were all in agreement with the primary particle diameter values found in TEM studies ($d_{TEM}$, Table 1). Nitrogen adsorption measurements (BET) revealed specific surface areas (SSAs) and particle sizes ($d_{BET}$), which were, again, in good agreement with the primary particle sizes $d_{XRD}$ and $d_{TEM}$, indicating only limited sinter-neck formation or particle aggregation (Table 1). The hydrodynamic diameters (Z-average) of all flame-made nanoparticles in water were comparable with values between 100 and 130 nm (Table 1). The citrate stabilized Au nanoparticles showed a Z-average close to their primary particle size and comparable to those of the FSP-synthesized nanoparticles. Organic residues on the nanoparticle surface originating from nanoparticle synthesis were estimated using thermogravimetric analysis (TGA). All nanoparticles showed surface residues of ≤2 wt% (weight percent). Only $TiO_2$ showed organic contents of up to 4 wt%, which could be further reduced by post-annealing. All particle types were well-dispersible in water.

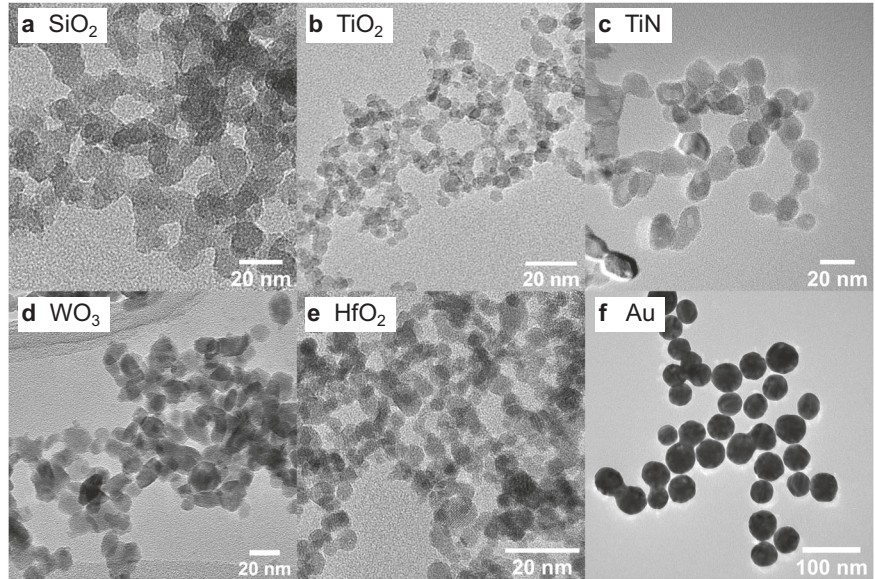

**Fig. 1 Nanoparticle morphology.** Representative TEM images (overall more than 200 nanoparticles and at least five regions of the sample analyzed for each particle type) of commercially available $SiO_2$ (**a**, A380, Evonik Industries AG, Germany), FSP synthesized metal oxide or nitride nanoparticles (**b–e**) and commercially available 50 nm Au nanoparticles (**f**).

**Table 1 Physicochemical properties of the nanoparticles used in this study.**

| Nano-particle | $d_{TEM}$ (nm) | $d_{XRD}$ (nm) | SSA (m²/g) | $d_{BET}$ (nm) | Z-average (nm) | Surface residues (wt%) |
|---|---|---|---|---|---|---|
| $SiO_2$ | N.A. | Amorphous | 343.6 ± 2.3 | 6.5 ± 1.0 | 113 ± 4 | 1–2 |
| $TiO_2$ | 5.2 ± 1.7 | 6.7(1)[t,a], 71% 3.2[t,r], 29% (partly amorphous) | 235 | 5.9 | 130 ± 4 | ~4" |
| TiN | 14.8 ± 5.5 | 15.6(12)[c] | 62.7 | 17.7 | 122 ± 6 | ~0.3 |
| $WO_3$ | 12.1 ± 4.5 | 9.6(1)[m, 40%] 4.2 [m(WO2.9), 27%] 13.4(1)[o, 15%] 8.5(1)[t, 18%] | 78.6 ± 0.7 | 10.6 ± 1.2 | 103 ± 19 | ~0.5 |
| $HfO_2$ | 4.9 ± 1.8 | 5.4(2)[m, 86%] 6.3(1)[o, 14%] | 114.0 ± 1.1 | 5.4 ± 0.6 | 130 ± 8 | ~2 |
| Au | 52 ± 6* | N.A. | 5.9* | 52.9 | 58 ± 1 | N.A. |

Values are displayed as the mean ± standard deviation (SD) or mean with last digits of the estimated SD (ESD) in brackets; XRD crystal phases: a: anatase, c: cubic, m: monoclinic, o: orthorhombic, r: rutile, t: tetragonal. *According to the analysis certificate of the supplier; "can be further reduced to <0.5% by post-annealing. The nanoparticle primary particle sizes ($d_{TEM}$) were extracted from TEM data by analyzing the diameter of $n = 211$ ($TiO_2$), $n = 247$ (TiN), $n = 195$ ($WO_3$), and $n = 275$ ($HfO_2$) nanoparticles.

**Cytocompatibility and cellular uptake.** All oxide nanoparticles were generally well tolerated by human sarcoma cells (HT1080). Sham-irradiation (0 Gy) experiments revealed lethal $LC_{50}$ values well above 160 μg/mL for all nanoparticles and exposure times of 5 days, except TiN, where the $LC_{50}$ was 157 μg/mL (Supplementary Fig 1). Cellular uptake was investigated for the aforementioned nanoparticles with electron microscopy and elemental analysis. Figure 2 shows scanning electron micrographs of HT1080 cells after 24 h of incubation with the indicated nanoparticles. Nanoparticles were taken up by cells, most likely via an endocytic pathway[30], and formed intracytoplasmic agglomerates, well in line with earlier studies of flame-made nanoparticles[31]. Few hundred nanometer up to micrometer-sized nanoparticle agglomerates were distributed within the cell cytoplasm (in vesicles or endosomes). In the >100 cells analyzed per nanoparticle type, no evidence for nanoparticle uptake into the nucleus was found, even though uptake overall, and nanoparticle accumulation in the nucleus, might be particle and cell type dependent[32,33]. Nanoparticle uptake was comparable for all types of oxides except for $WO_3$ nanoparticles, for which only very few nanoparticle agglomerates were found intracellularly (Fig. 2d).

Elemental analysis based on inductively coupled plasma mass spectrometry (ICP MS) was employed to quantify the metal mass

normalized to the cell number. For nanoparticle concentrations of 80–160 μg/mL, an uptake of ~1 ng of metal mass per HT1080 cell was found for most particles. ICP-MS results confirmed the observations from TEM images, and revealed that intracellular metal concentrations were two orders of magnitude lower for tungsten than for all other metals (Fig. 2g and Supplementary Fig. 2). Additional investigations confirmed the dissolution of $WO_3$ nanoparticles in acidic, lysosome-mimicking conditions (Supplementary Fig. 3).

**Physical contributions to dose enhancement.** Following the physicochemical characterization of the synthesized nanoparticle radio-enhancer candidates and cellular uptake studies, we performed Monte Carlo simulations using TOPAS/Geant4 to estimate the contribution that can be expected from physical dose enhancement as a function of core material and type of irradiation (150 kVp photons, 6 MV photons or 100 MeV protons). We investigated the dose deposition within and around an individual, nanoparticle filled vesicle, as well as within the cytoplasm and nucleus of a single cell. We extracted physical dose-enhancement factors (DEFs) by building the ratios of the dose scored to the cytoplasm, nucleus, vesicle, or water shells, respectively, in the presence of the nanoparticles to that with no

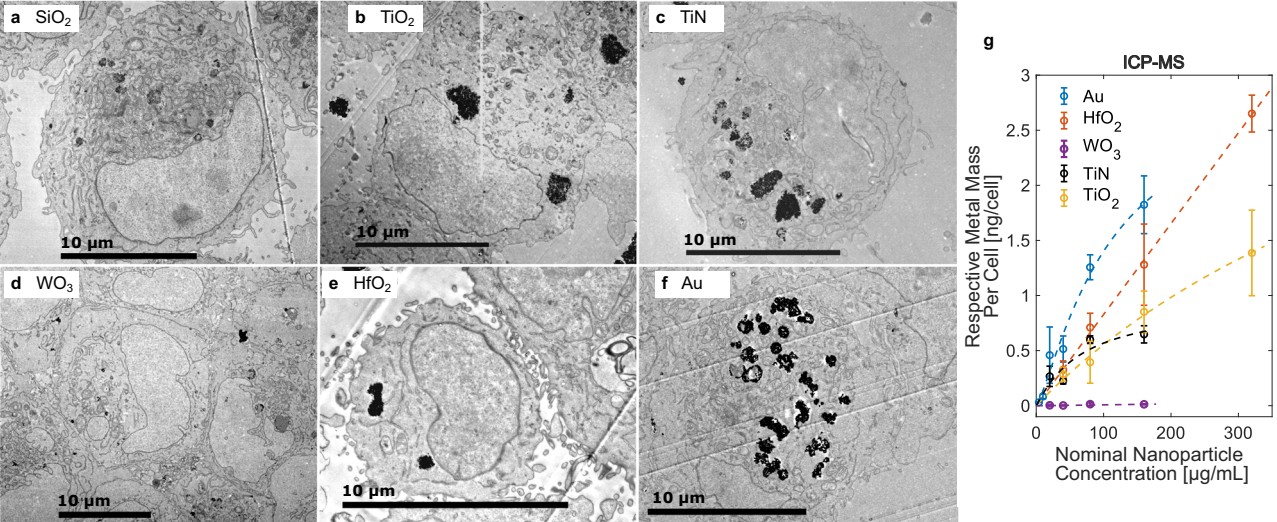

**Fig. 2 Cellular uptake of nanoparticles.** Scanning transmission electron (STEM) micrographs of HT1080 cells exposed to 80 µg/mL SiO$_2$ (**a**), TiO$_2$ (**b**), TiN (**c**), WO$_3$ (**d**), HfO$_2$ (**e**), and Au (**f**) nanoparticles for 24 h. Intracellular and cell-associated metal mass normalized to cell number as quantified by ICP-MS (**g**). Data in **g** expressed as mean ± SD from $n = 4$ biological repeats per data point examined over 2 independent ICP experiments.

nanoparticles (water). The geometries were built to match cellular uptake scenarios as closely as possible, with ~400 nm nanoparticle agglomerates distributed only within the cytosol (see also Fig. 2)[31]. As nanoparticle uptake into the nucleus was not observed experimentally, it was considered negligible also for the simulations.

**Nanoparticle-filled vesicle model—nanoscopic physical dose enhancement.** Nanoscopic dose enhancement stems from secondary electron emission after ionization of a nanoparticle, with Auger electrons contributing to low range (~10 nm) and electrons of higher energy (Photo- or Compton electrons) contributing to micrometer range enhancements[34,35]. Figure 3a, b show the simulated dose enhancements for different nanoparticles within a nanoparticle-filled vesicle with 200 nm radius and within 50 nm water shells surrounding the vesicle. A DEF value >1 indicated additional dose deposition by nanoparticles compared to water, while a DEF value equal to 1 meant that no additional dose deposition was observed. For lower energy X-rays (150 kVp source), the highest dose deposition and a clear impact of atomic number were observed. This is in line with the different mass energy absorption cross-sections caused by the photoelectric effect for high-$Z$ metals. The dose-enhancement factors within a nanoparticle-filled vesicle reached values of DEF = 30–40 for Au nanoparticles and DEF = 10–20 for HfO$_2$ and WO$_3$ nanoparticles at the highest reached nanoparticle content of 32.4 vol% (volume percent) in the vesicle (Fig. 3a). This packing fraction is also reasonable for biological scenarios. For instance, nanoparticle volume fractions of 35 ± 16% per vesicle have been reported in cells for 30-nm-sized Au nanoparticles[36], and exposure conditions similar to the ones used in our study. Low-$Z$ nanoparticles, such as TiO$_2$, TiN, and SiO$_2$ showed no nanoscopic dose increase at all. The dose enhancement decay from the filled vesicle surface followed a 1/$r$–type decay, and the DEF converged to DEF = 1 within one micrometer of the cytoplasm (Fig. 3b). For MV X-rays, dose enhancement within and around nanoparticle-filled vesicles was only found for Au nanoparticles (Supplementary Fig. 4). This enhancement was found to be even more localized, converging to DEF = 1 within 100 nm from the vesicle surface. Nanoscopic physical enhancement of proton irradiation was negligible for all nanoparticles (Supplementary Fig. 4b).

**Cell model—microscopic physical dose enhancement.** The total DEF within the cytoplasm or nucleus increased linearly with increasing nanoparticle filling. This is in line with the calculation of the macroscopic DEF

$$\mathrm{DEF}_{\mathrm{macroscopic}} = 1 + f_Z \cdot \left(\frac{\mu_{\mathrm{en}}(E)}{\rho}\right)_Z \bigg/ \left(\frac{\mu_{\mathrm{en}}(E)}{\rho}\right)_{\mathrm{H}_2\mathrm{O}} \quad (1)$$

where $f_Z$ is the atomic number ($Z$) mass fraction in the system and $\mu_{\mathrm{en}}(E)/\rho$ the mass energy-absorption coefficient at a monoenergetic photon energy, $E$[37]. Mass and volume fraction ($f_{\mathrm{Vol}}$) are related via a constant density ratio of the materials. We therefore used a linear fit to describe the dose-enhancement efficiency of a nanoparticle (NP) per cellular nanoparticle volume fraction, $\chi_{\mathrm{NP}}$, within the cytoplasm and nucleus, with the relationship $\chi_{\mathrm{NP}} = (\mathrm{DEF} - 1)/f_{\mathrm{Vol,NP}}$ (Supplementary Fig. 5).

It is evident that for Au, HfO$_2$, and WO$_3$ considerable additional doses were deposited in the cytoplasm and in the nucleus, but only at kV energies. In the cytoplasm, the DEF reached values of approximately 10.5, 5.0, and 3.7, while in the nucleus it reached 5.7, 2.9, and 2.3 per nanoparticle volume fraction percent of Au, HfO$_2$, or WO$_3$, respectively. Using the 6 MV X-ray spectrum, the dose-enhancement efficiency per nanoparticle volume fraction was roughly 10 times reduced, compared to the 150 kVp source. We found DEF values of 1.9 and 1.4 for Au, 1.3 and 1.2 for HfO$_2$ and 1.2 and 1.1 for WO$_3$ nanoparticles per volume percent in the cytoplasm and nucleus, respectively. For low-$Z$ nanoparticles such as TiO$_2$, TiN, or SiO$_2$, no physical nanoparticle enhancement was detected, since no considerable additional dose was deposited in the cytoplasm, nucleus, or nanoparticle-filled vesicle for any type of irradiation source. No dose-enhancement effects were found for any of the investigated nanoparticles for the proton beam source (DEF = 1).

While no comprehensive study is available in the literature, data for Au nanoparticles are available from Rudek et al. for a comparable cell model. They reported DEFs of 2 and 3.5 in the nucleus and cytoplasm, respectively, for 100 kV monoenergetic X-rays and 5 wt% randomly distributed Au nanoparticles[38]. In our study these values were 2.2 (nucleus) and 3.5 (cytoplasm) for 0.26 vol% Au, which relates to 5 wt% Au, and are thus well in line with their data. Furthermore, we agree with their finding, that

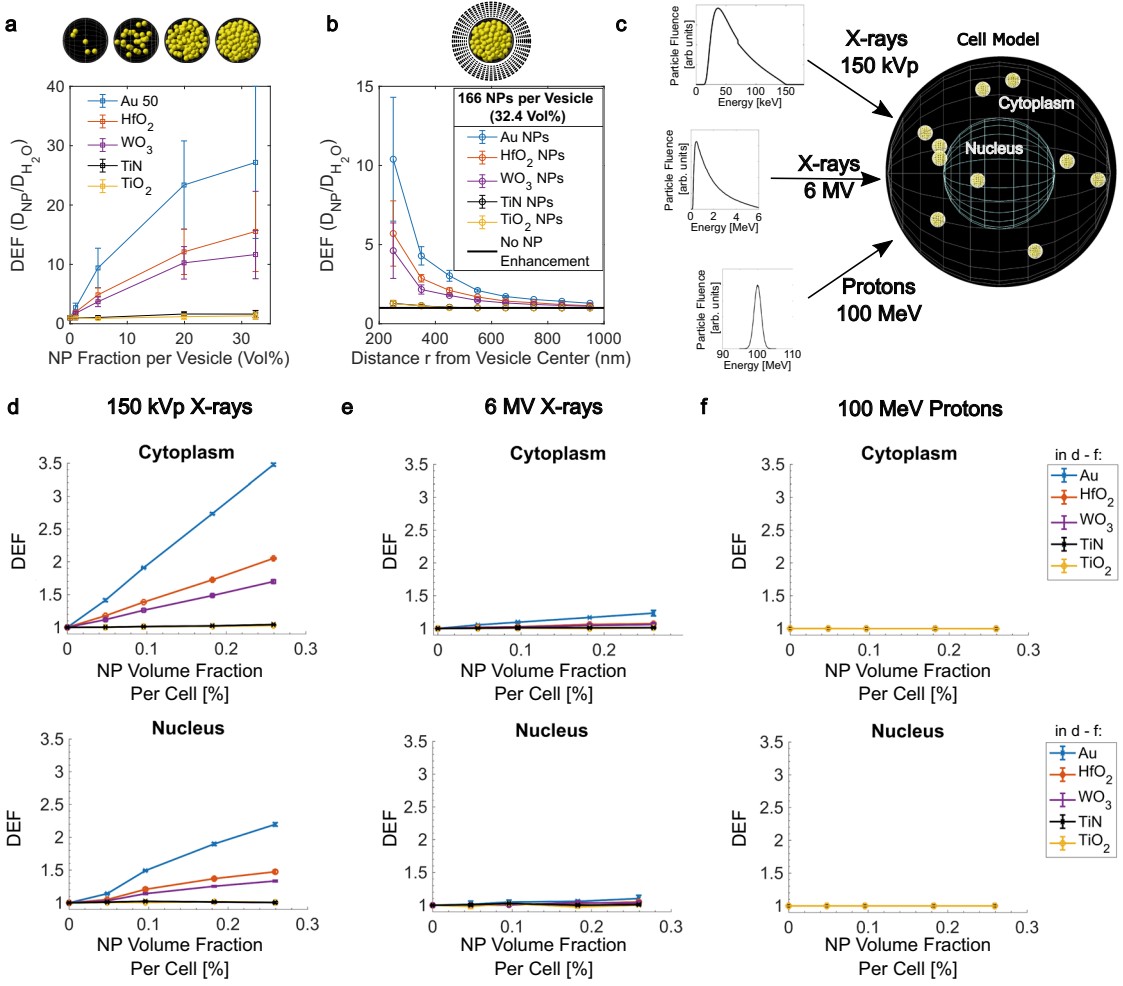

**Fig. 3 Physical dose enhancement factors (DEFs) for 50-nm-sized TiO₂, WO₃, HfO₂ and Au nanoparticles assessed using Monte Carlo simulations.** DEF inside a 400 nm water vesicle filled with different amounts of nanoparticles upon irradiation with 150 kVp X-rays (**a**). DEF in 100 nm shells around a 400 nm-sized nanoparticle filled with 166 (=32.4 vol%) nanoparticles upon irradiation with 150 kVp X-rays (**b**). Schematic illustration of the cell model consisting of a central nucleus and nanoparticle-filled vesicles inside the cytoplasm (no nanoparticles in the nucleus) and the particle influence for the different photon and proton beam sources used (**c**). DEF in the cytoplasm and nucleus for 150 kVp X-rays (**d**), 6 MV X-rays (**e**) and 100 MeV protons (**f**). Data in **a**, **b** and **d**–**f** given as mean ± SD from $n = 3$ simulation experiments. Source Data are provided as a Source Data file.

intracellular physical dose enhancement is rather negligible for protons.

**Chemical contributions to dose enhancement.** Locally enhanced physical dose effects and an increased ionization of oxygen-containing molecules in the vicinity of the nanoparticle can lead to generation of reactive oxygen species (ROS)[1,9]. Additionally, catalytic reactions on the nanoparticle surface can lead to enhanced radiolysis and ROS formation by lowering the ionization potentials of molecules at the nanoparticle–liquid interface or by electron donor processes[1,39]. In this way, electrons with lower energies than typically necessary can also lead to water ionization. We thus investigated the enhanced ROS generation by nanoparticles under irradiation using the well-known 2′,7′-dichlorodihydrofluorescein diacetate (H₂DCF-DA) assay. This assay has one of the highest reactivities among the established ROS assays for generated •OH radicals and is therefore a reasonable choice for quantification of ROS[40]. While SiO₂ and TiN nanoparticles showed no increase in ROS formation (DEF_ROS ~ 1), TiO₂, WO₃ and HfO₂ nanoparticles showed increased ROS formation with increasing particle concentration (DEF_ROS > 1) under all types of ionizing irradiation (Fig. 4). ROS formation

under X-ray irradiation was generally higher than that under proton irradiation (Fig. 4c–e). Fitting of a linear regression for ROS enhancement versus nanoparticle surface area concentration revealed ROS enhancement efficiencies that decreased in the orders: WO₃ > TiO₂ > HfO₂ under kV X-ray irradiation and TiO₂, WO₃ > HfO₂ under MV X-ray and proton irradiation (Fig. 4f).

For these measurements of chemical dose enhancement, the total metal content at all nanoparticle concentrations tested was below 0.3 wt% or below 0.05 vol%. At such low percentages, physical dose enhancement is negligible, especially for nanoparticles other than Au and for MV X-rays or protons (DEF < 1.2, see Fig. 3d–f). The ROS dose enhancement (DEF_ROS) was at least 10–100 times higher than the physical dose-enhancement effect per nanoparticle volume percent (Supplementary Fig. 5). Therefore, we concluded that the nanoparticle effects observed in this assay were based primarily on catalytic surface effects. To exclude effects from synthesis-related organic residues on the surface of TiO₂ nanoparticles (Table 1), we conducted an annealing study using temperatures of up to 500 °C to remove such residues. DEF_ROS remained unaffected by annealing (Supplementary Fig. 6), and thus, we concluded that surface organic residues had no measurable influence on ROS generation under irradiation in our setting.

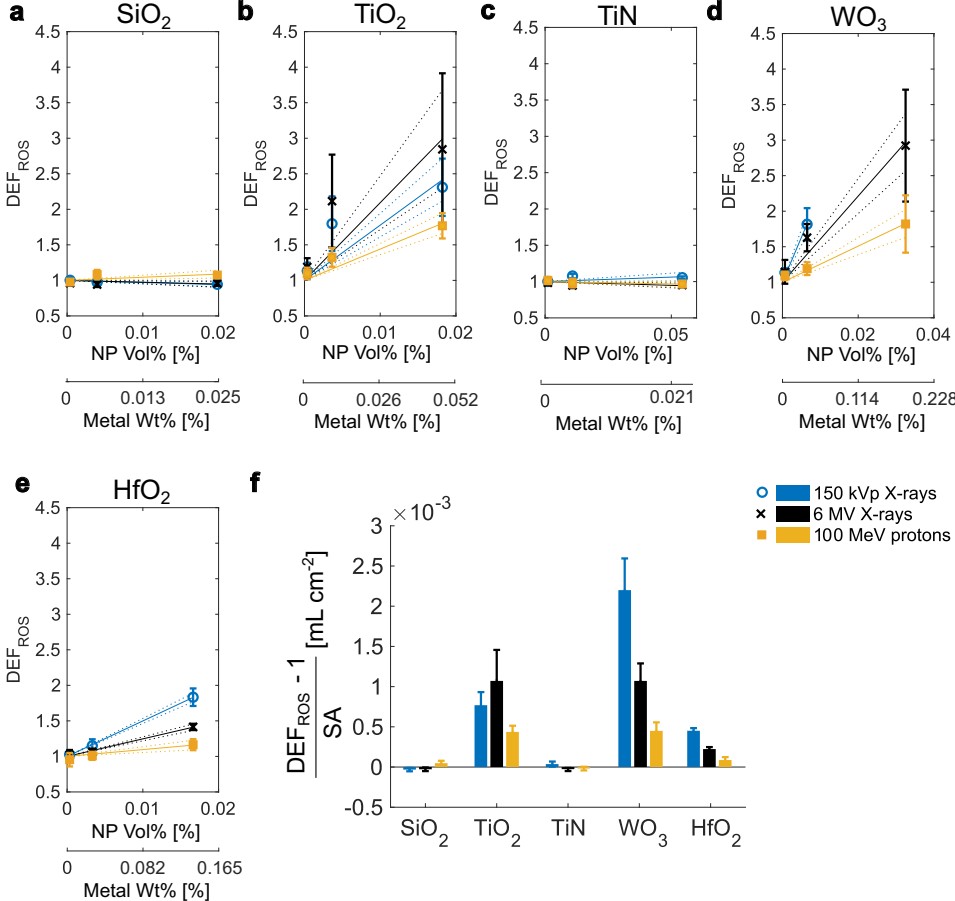

**Fig. 4 Reactive oxygen species (ROS) generation in nanoparticle suspensions.** Chemical dose-enhancement factors (DEF$_{ROS}$) for SiO$_2$ (**a**), TiO$_2$ (**b**), TiN (**c**), WO$_3$ (**d**), and HfO$_2$ (**e**) as assessed using the acellular H$_2$DCF-DA assay under 150 kVp and 6 MV photon, as well as 100 MeV proton irradiation. The exposed nanoparticle surface areas were kept constant (36.8, 368, and 1840 cm$^2$/mL), thereby giving different volume or metal mass concentrations (x-axes) for each nanoparticle. Data in **a–e** expressed as mean ± SD for $n = 6$ samples examined over 2 independent DCF experiments. Straight and dotted lines show slope and 95% confidence level prediction bounds of linear fit, respectively. Bars and error bars in **f** represent the slopes and 95% confidence interval from the linear regression response for DEF$_{ROS}$ per surface area concentration of each nanoparticle and for each irradiation condition ($n = 18$ sample points per regression), respectively.

Catalytic processes have previously been postulated to explain the chemical enhancement, which was found to be several magnitudes higher (e.g., up to 2000 times)[41] than what could be expected from purely absorption-related processes during irradiation of nanoparticle suspensions[39,41]. One such proposed catalytic mechanism involves the formation of a structured water layer around nanoparticles leading to weakened H–OH bonds and thereby to more efficient water radiolysis in the vicinity of nanoparticles[39]. Such a mechanism would be nanoparticle material independent. In our study, ROS amplification for comparable surface exposures was dependent on the material composition, indicating that the catalytic process at hand rather involves a charge transfer process specific to the nanoparticle composition. Semiconductor particles, with or without band-gap engineering, have been shown to possess photocatalytic properties, which can be harnessed, for example, to decompose organic molecules, such as methylene blue or ethylene[42–44]. TiO$_2$ and WO$_3$ are thought to be particularly effective for production of •OH radicals and photodegradation of organic molecules due to the relative position of their charge and valence band potentials[45]. Therefore, we concluded that ROS generation with TiO$_2$ and WO$_3$ was superior to that with HfO$_2$ nanoparticles for all irradiation conditions (kV/MV photons and protons), due to their more beneficial energy band potentials.

For HfO$_2$ nanoparticles, a clear impact of the radiation type on ROS enhancement was observed with enhancement ratios scaling roughly as 1:2:4 for (MeV protons):(MV X-rays):(kV X-rays) (Fig. 4e, f). The different catalytic efficiencies for kV versus MV X-rays may be related to higher absorption cross-sections at kV energies, suggesting also a higher excitation probability at such energies. Nanoparticle excitation leading to electron–hole pair generation is the fundamental process behind photocatalytic surface reactions[46]. We thus concluded that (i) proton irradiation led to lower nanoparticle excitation compared to X-rays, and (ii) the energy spectrum (150 kVp vs. 6 MV) of the X-rays had an influence on the electron–hole pair generation for high-Z nanoparticles (HfO$_2$ and WO$_3$) but not for TiO$_2$ nanoparticles. Au nanoparticles also possess the ability to generate ROS under X-ray[41,47,48] or proton[49] irradiation. Using 150 kVp X-rays and acellular ROS assays, Au NPs were found to generate additional •OH radicals[48]. Physical and catalytic surface effects as well as an inverse size dependence have been suggested to play roles in the chemical enhancement by Au NPs[41,47,50,51]. Due to strong assay interference we were, however, not able to confirm these results for Au nanoparticles in our setting.

Taken together, the above investigations indicated a strong role for radio-catalytic processes in the chemical stage of nanoparticle radio-enhancement.

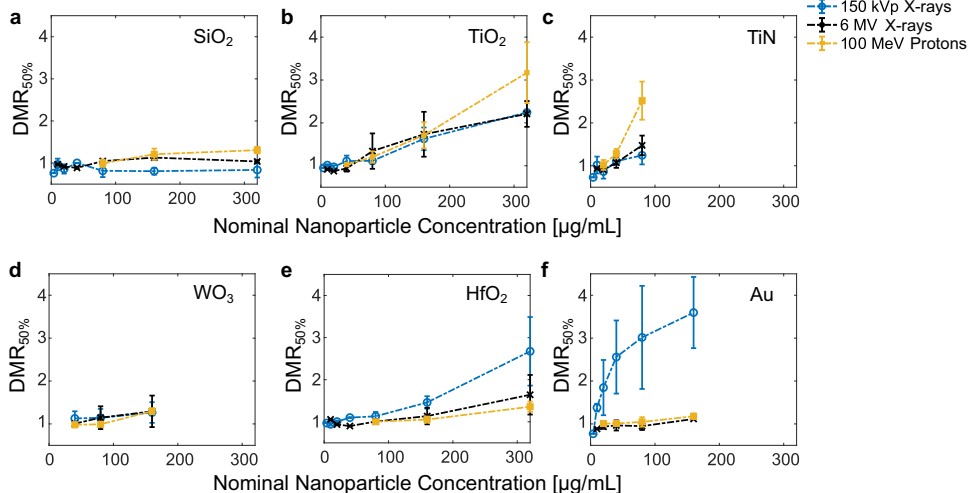

**Fig. 5 Nanoparticle radioenhancement in vitro.** Dose modifying ratios at 50% cell survival ($DMR_{50\%}$) observed in vitro with different nominal nanoparticle concentrations of $SiO_2$ (**a**), $TiO_2$ (**b**), TiN (**c**), $WO_3$ (**d**), $HfO_2$ (**e**), and Au (**f**) nanoparticles in HT1080 cells; data extracted from survival curve with $n = 15$ independent biological samples (3 samples per X-ray dose) and expressed as mean ± SD from at least 2 independent cell batches.

**In vitro effects of dose enhancement**. In addition to physical and chemical contributions to dose enhancement, nanoparticle dose-enhancement effects in cell relatively radioresistant human soft tissue sarcoma cells (HT1080) were quantitatively assessed from cell survival curves (Supplementary Fig. 7). The dose-enhancement effects observed in vitro was evaluated by calculating the dose modifying ratio at 50% cell survival ($DMR_{50\%}$), which is the ratio of radiation doses that led to the same effect without versus with nanoparticles (Supplementary Fig. 8). Figure 5 shows the DMRs extracted for all nanoparticles and for the different irradiation sources used. Apart from $SiO_2$ nanoparticles, which served as a negative control (no physical or chemical enhancement under irradiation) and for which, expectedly, no considerable in vitro dose enhancement was found, all nanoparticles showed increased DMRs with increasing nanoparticle concentration. The highest dose-enhancement effects, up to 300%, were found for Au nanoparticles under kV irradiation. Interestingly, under MV X-ray or proton irradiation, the dose-enhancement effect for Au nanoparticles dropped drastically to values just above baseline (<20% dose enhancement). This can be attributed exclusively to radiation beam effects as the only variable, since the nanoparticles and cell model were exactly the same for all three irradiation conditions (kV, MV X-rays, and protons). $HfO_2$ showed dose-enhancement effects up to 200%, 70% and 40% under kV X-ray, MV X-ray and proton irradiation, respectively. Note that $HfO_2$ nanoparticles showed considerably higher enhancement effects than Au during MV X-ray and proton irradiation, hence possibly justifying their success in clinics. In the case of $WO_3$, dose-enhancement effects of up to 50% were detected regardless of the type of ionizing irradiation used. Interestingly, for $TiO_2$ nanoparticles, the dose-enhancement effects observed were comparable for kV and MV X-ray treatments and reached values up to 150%. Under proton irradiation, enhancement effects of up to 290% were observed for $TiO_2$ at the highest dose. For TiN nanoparticles, up to 50% dose enhancement was detected for subtoxic doses during kV and MV X-ray irradiation. The strongest effect by TiN nanoparticles was found with proton irradiation, with enhancements of up to 200%.

There is a scarcity of studies comparing different irradiation sources on the same setup. Literature data exist exclusively for Au nanoparticles. Available studies for Au nanoparticles hint at strongly attenuated radioenhancement for MV X-ray and proton irradiation conditions, compared to that for kV X-rays. Our results are in line with those of Smith et al. (2015), who showed

dose enhancements at 5 Gy of ~60% for 80 kVp X-rays, 10% for 6 MV X-rays and <5% for 150 MeV protons using alanine wax impregnated with 3 wt% of 5 nm Au nanoparticles[29]. Similarly, Chithrani et al. (2010) found enhancement effects for 50 nm Au nanoparticles of 66% with 105 kVp, which dropped to 17% with 6 MV X-ray irradiation using in vitro clonogenic assays[52]. Based on the literature, it is evident, that reported in vitro dose-enhancements observed for Au nanoparticles are most commonly in the range of 0–100% for low energy (kV) X-rays and well below 50% for MV X-rays[12,53]. Our results for 50 nm Au nanoparticles validate these observations and showed a very pronounced drop in radioenhancement efficacy in clinically relevant MV radiotherapy and proton therapy settings. These results further indicate the clear limitations of 50 nm Au nanoparticles and highlight the urgent need for high-performance nanoparticle radio-enhancers with rational core material selection and particle design for clinically relevant high-energy irradiation. These findings also validate the clinical use of $HfO_2$ nanoparticle radio-enhancers and indicate even stronger benefits with catalytically active materials, such as $TiO_2$, in combination with a high surface area per volume.

**Unraveling the dominant mechanisms under different irradiation conditions**. In order to gain mechanistic insights, we correlated the in vitro dose-enhancement effects to the nanoparticle volume fraction (Supplementary Fig. 9) to compare relative nanoparticle performance and enable contextualization of our physical and chemical dose enhancement results. For kV X-ray irradiation, the relative trend in DMR can be very well predicted with the physical DEF simulations, well in line with earlier findings[31]. For this energy regime, the atomic number plays a crucial role contributing to physical effects (i.e. increased energy deposition within the cytoplasm and nucleus). Consequently, for kV photons, Au was the most efficient radio-enhancer, followed by $HfO_2$, followed by $TiO_2$ and TiN.

For MV photon beam irradiations, the atomic number dominance is overruled by the chemical enhancement activity. Highest dose enhancements under 6 MV X-ray irradiation were reached for $TiO_2$ and $HfO_2$. Strikingly, our results indicated that 50 nm Au nanoparticles were less efficient than the oxides, even at comparably higher volume fractions.

For proton irradiation, $HfO_2$ and Au showed lowest radiation enhancement efficacies. Dose-enhancement efficacy of $HfO_2$ nanoparticles with protons was 5 and 2 times lower compared

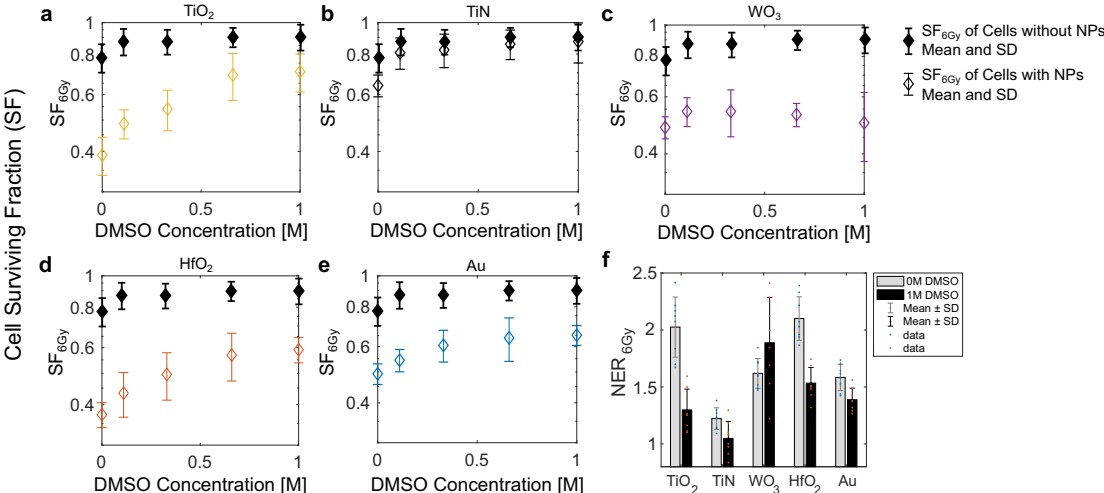

**Fig. 6 Radical quenching.** Impact of DMSO (hydroxyl radical scavenger) on the surviving fraction (SF) of HT1080 cells with and without nanoparticles after irradiation with 6 Gy of 150 kVp X-rays. Cells have been pre-incubated with 160 µg/mL TiO$_2$ (**a**), 80 µg/mL TiN (**b**), 160 µg/mL WO$_3$ (**c**), 320 µg/mL HfO$_2$ (**d**), and 40 µg/mL Au nanoparticles (**e**) for 24 h before DMSO and X-ray treatment. Monodose nanoparticle enhancement ratio (NER) at 6 Gy without and with 1 M DMSO (**f**). Data in **a**–**f** expressed as mean ± SD from $n = 8$ biological replicates examined over 2 independent CellTiter Glo experiments for each nanoparticle type or vehicle.

to the ones with kV and MV photons, respectively, in line with chemical enhancement results (see Supplementary Fig. 4). Proton dose-enhancement efficacy of Au nanoparticles was comparable to the one with MV X-ray irradiation and ~20 times lower than the one with kV X-ray irradiation. Surprisingly, titanium-based materials outperformed the other materials showing higher proton enhancement than suggested by the physical or catalytic mechanisms presented here. This strong dose enhancement was likely caused by another mechanism specific to proton interactions with those elements. When natural titanium ($^{48}$Ti) is bombarded with protons it can undergo a nuclear reaction via $^{48}$Ti(p,x)$^{48}$V to produce the positron emitting radioisotope $^{48}$V[54,55]. Similarly, proton bombardment on nitrogen targets can yield the positron emitters $^{13}$N or $^{11}$C via $^{14}$N(p,pn)$^{13}$N or $^{14}$N(p,α)$^{11}$C reactions[56]. While positron emitters, as opposed to alpha emitters, are generally useful for diagnostics using positron emission tomography, recent studies have proposed that positrons might also be useful for cancer therapy[57]. Whether nuclear reactions or other biological mechanisms under proton irradiation enhanced Ti- and N-containing materials more than under X-rays in our case should be investigated further. Literature on proton enhancement using TiO$_2$ or TiN nanoparticles is largely missing.

**Deciphering catalytic contributions.** Ionizing radiation-induced cell damage is the result of cooperative contributions from indirect and direct cell stress mechanisms, including ROS generation, DNA damage, and subcellular organelle damage and autophagy[58]. Indirect actions are thought to be mediated mostly by hydroxyl radicals (•OH)[59] and are estimated to contribute up to 90% to DNA damage or cell lethality[60,61]. For charged particles such as protons, a slightly more significant contribution of direct action to cell killing has been described, since they are more densely ionizing than X-rays[27]. •OH radicals can be quenched efficiently using dimethylsulfoxide (DMSO) to protect cells from X-ray and proton damage[27,60,62]. A maximum degree of protection was found for concentrations of approximately 1 M DMSO[61]. Here, we investigated the radioenhancement mechanism of nanoparticles by using DMSO in different concentrations to quench hydroxyl-mediated damage mechanisms.

Figure 6 shows the effects of radical quenching during kV X-ray irradiation. Radiation treatment in combination with DMSO and/or nanoparticles revealed very interesting effects. Nanoparticle concentrations were chosen such that radioenhancement effects were expected, while sham irradiation (0 Gy) effects were negligible (Supplementary Fig. 10). Indeed, all nanoparticles revealed dose-enhancement effects, which can be expressed as monodose nanoparticle radiation enhancement ratio (NER or RER[63]). We give this ratio as

$$NER = \frac{SF_{6Gy,no\,NP}}{SF_{6Gy,NP}} \quad (2)$$

where SF$_{6Gy}$ is the normalized cell surviving fraction at a dose of 6 Gy. For WO$_3$ nanoparticles, the NER was ~1.6 with or without DMSO (Fig. 6f). For all other nanoparticles, increasing DMSO concentrations decreased the nanoparticle radiation-enhancement effect until effect saturation occurred. The complete protection from nanoparticle radioenhancement effects was observed with TiN nanoparticles and 0.1 M DMSO or more, showing a NER of 1.3 without DMSO and a NER close to 1.0 with DMSO. For TiO$_2$, HfO$_2$ and Au nanoparticles, the NERs decreased from 2.0, 2.1, and 1.6 (no DMSO) to 1.3, 1.5, and 1.4 at >0.5 M DMSO, respectively. In other words, addition of DMSO suppressed the total nanoparticle radiation-enhancement effects by up to 73% for TiO$_2$, 52% for HfO$_2$ and 34% for Au nanoparticles (Supplementary Fig. 11). These percentages reflect the cell damage enhancement mediated by •OH radicals. Short treatment with up to 1 M DMSO alone did not show negative effects on cell growth (Supplementary Fig. 10a). For nanoparticle-free samples, cell survival fractions of 70–80% were found after 6 Gy X-ray treatment, which were increased to 90% in presence of DMSO at concentrations of 0.1 M or higher (Fig. 6). These findings are in agreement with previous reports which amounted the percentage of indirect action to 63–89%[60–62,64]. The remaining, not DMSO suppressed nanoparticle dose enhancement must stem from other physical, chemical, or biological damage mechanisms.

TiO$_2$ has been shown to activate cellular stress and reduce metabolic capacity[65], elevate the levels of intracellular ROS, inflammation-related genes and apoptosis, and reduce GSH levels[66]. Increased DNA damage and micronucleus formation, a probable mechanism of genotoxicity, has also been reported[67].

Nevertheless, our results indicated that most (>73%) of the nanoparticle enhancement was due to production of ROS, especially •OH, during irradiation with ionizing radiation. Youkhana et al. (2017) also attributed the in vitro dose enhancement of TiO₂ to increased ROS generation based on aqueous DCFDA results[68]. Here, we were able to show conclusively that this oxidative stress increased during irradiation most likely due to the catalytic surface effects of TiO₂, since no physical dose enhancement was found in our nanoscopic and microscopic models.

DMSO addition only partly attenuated nanoparticle enhancement by high-Z nanoparticles HfO₂ and Au. Thus, direct DNA damage might occur at kV X-ray irradiation for high-Z nanoparticles, which is consistent with our simulations of microscopic dose-deposition within the cell nucleus. Additionally, •OH radical formation in the cytosol might then be a combination of physical and surface catalytic processes for these high-Z materials at kV X-ray energies. Interestingly, cytoplasmic processes leading to the disruption of organelles, such as mitochondria or lysosomes, may play a major role in nanoparticle-mediated radioenhancement[69,70]. Most recently, it was also shown that even very low concentration of 10 nm Au nanoparticles can have an effect on cell cycle phase, the proportion of radiosensitive G2 cells as well as the DSB repair kinetics[71]. Thus, the nanoparticle-mediated radiation response is complex with sensitization of cancer cells as well as dose enhancement both contributing to the overall response.

The only nanomaterial for which DMSO showed no effect on the radioenhancement efficacy was WO₃. Since in our case WO₃ nanoparticles dissolved prior to irradiation, their action must be governed by low concentrations of free tungsten ions. Using soluble sodium tungstate, it has been shown that tungsten can alter cell cycle progression, stimulate cytokine profiles and increase DNA damage and apoptosis[72–74]. Furthermore, it has been reported that tungstate modulates the activation of the central double strand break (DSB) signaling kinase ATM and that sensitized cells to DNA DSB-inducing agents, such as ionizing radiation[75]. Such a biological sensitization mechanism could explain our findings that (dissolved) WO₃ nanoparticles sensitized HT1080 cells under X-ray or proton irradiation to similar extents without increasing oxidative (•OH radical) stress.

We postulate the following mechanisms of action for the nanomaterials used in this study:

*Silicon dioxide*: SiO₂ is a biologically relatively inert nanomaterial that does not produce relevant ROS or cell sensitization or lead to observable dose enhancement effects. It can be used as a control for further radioenhancement studies.

*Titanium nitride*: No physical dose enhancement in the cytoplasm or nucleus was found for TiN nanoparticles. TiN might reduce the antioxidant capacity and sensitize cancer cells to ionizing radiation. With low concentrations of DMSO, the dose enhancement effect was diminished completely. Under proton irradiation, however, nuclear effects of ⁴⁸Ti and ¹⁴N atoms may play additional roles leading to increased proton dose enhancements observed in vitro.

*Titanium dioxide*: The in vitro dose enhancement by TiO₂ nanoparticles under X-ray irradiation did not show an energy dependence. Any physical dose enhancement in vesicles, cytoplasm or nucleus can be excluded. Ionizing radiation generated cellular oxidative stress via catalytic reactions, which were higher under X-ray than under proton irradiation. An additional biological effect (<30%) might play a significant role, since most but not all dose enhancement was quenched by DMSO. High availability of nanoparticles per cell were necessary to show high dose enhancement effects. This was allowed by the

low toxicity profile. Under proton irradiation, nuclear effects may play an additional role in further enhancing cell damage.

*Tungsten oxide*: WO₃ nanoparticles can generate physical dose enhancement under kV X-ray irradiation and surface catalytic ROS under all ionizing irradiation sources used. However, nanoparticle dissolution—likely in lysosomal compartments—led to low tungsten content per cell and to a rather biological than oxidative sensitization of the cells toward X-ray and proton irradiation.

*Hafnium dioxide*: HfO₂ nanoparticles enhanced the dose within the cell nucleus and enhanced cellular oxidative stress via dose deposition in the cytoplasm as well as via surface catalytic reactions under kV X-rays. Catalytic processes were reduced under MV X-ray irradiation and were lowest under proton irradiation. In addition to a very low toxicity profile, HfO₂ nanoparticles showed proof of in vitro dose enhancement governed by physical and chemical phenomena under photon and proton irradiation.

*Gold*: 50 nm Au nanoparticles showed high dose enhancement effects at kV energies, which can be justified partly (<35%) by indirect (•OH radical mediated) and mostly by direct (dose to nucleus or cytoplasm) dose enhancement effects. These effects were strongly reduced with ionizing radiation at MV energies. The applicable nanoparticle surface of a 50 nm nanoparticle is 100 times less than that of a 5 nm nanoparticle, which reduces possible dose enhancement effects based on catalytic surface processes compared to all other nanoparticles.

Our study comprehensively investigated the physical, chemical and in vitro dose-enhancement effects of low-Z (SiO₂, TiO₂, and TiN) and high-Z (WO₃, HfO₂, and Au) nanoparticles during irradiation with kV and MV photons and protons in directly comparable settings. From this, valuable general findings can be extracted to inform rationalized nanoparticle radioenhancers design for the respective irradiation settings and treatment modalities. We conclusively showed that dose enhancement under kV X-ray irradiation is dominated by physical dose-enhancement effects, which can lead to enhanced dose deposition inside the cell's cytoplasm and nucleus. In vitro radioenhancement was highest for high-Z nanoparticles (Au). However, these physical dose-enhancement effects were strongly attenuated for high-energy photons and protons. In clinically most relevant MV photon and proton irradiation conditions, the catalytic activity of nanoparticles becomes decisively important, and the atomic number of the core material plays only a minor role. In order to take full advantage of surface catalytic ROS generation for dose enhancement, the accessible surfaces of radio-enhancers must be maximized. Interestingly, titanium- and nitrogen-containing nanoparticles might offer additional advantages for proton-enhancement therapy due to possible nuclear effects.

Taken together, the results of this comprehensive study allow extraction of important general design principles for nanoparticle radioenhancers for kV and MV photon and proton therapies based on quantitative and comparative data. While nanoparticle uptake, cellular toxicity, and radio-enhancement are cell-line dependent, previous work on different cancer cell lines has indicated that the relative trends in radioenhancer effectiveness hold true, albeit with slightly different absolute values[31]. Future investigations should focus on further exploration of the materials design space based on such comparative, well-standardized settings, and validation of the radioenhancement performance in animal models and with different cell types. Potential effects of nanomaterial surface functionalization on dose-enhancement should be investigated carefully, including potential ROS quenching by antioxidant molecules (such as dopamine), as well as potentially synergistic effects leading to augmented ROS generation, e.g. by porphyrins. Eventually, the effective nanoparticle concentration reached in the

cancer cells will govern dose enhancement. While experimental research in mouse models on intravenously injected nanoparticles has indicated that only a small fraction of the injected nanoparticles may accumulate inside cancer cells, intratumoral administration of $HfO_2$ nanoparticles (NBTXR3) appears to partially overcome the delivery problem and shows convincing therapeutic effectiveness in preclinical and clinical settings. This effectiveness may, however, be even further improved by optimizing radioenhancer designs, in parts based on the insights provided by this work.

## Methods

**Monte Carlo simulations**. To estimate the impact of nanoparticles on energy deposition, physical stage simulations were performed using the TOPAS-nBio[76,77] extension and the TOPAS[78,79] toolkit, which is based on the Geant4[80] Monte Carlo simulation system. We used the default Geant4-DNA track structure physics list[81] for the simulation of particle transport within the cell including the full Auger de-excitation cascade process[35] with fluorescence, Auger electron production, and particle-induced X-ray emission turned on. The energy range minimum was set to 10 eV. Since this physics module supports only transport in $H_2O$, we used the condensed-history EMStandardOption4 physics list in all nanoparticle regions, including nanoparticles made out of water, and specified a 1 nm range cut for all particles. The dose-enhancement factor (DEF) was defined as the ratio of the dose scored to the cytosol, nucleus, or to water shells in the presence of the nanoparticles of interest to that with water nanoparticles. Simulations using X-rays were performed on a MacOS Catalina distribution using TOPAS version 3.5 using triplicates of $10^8$ histories. The X-ray spectrum of the 150 kVp source was simulated using the XRayGUI (Version 1.4.2.0, BAM Federal Institute for Materials Research and Testing, Berlin, Germany) matching experimental conditions as closely as possible. For the X-ray spectrum of the 6 MV source, already published particle fluence within a water phantom from a Varian Linac was extracted from Choi et al. (2019)[82]. Proton simulations with a 100 ($\pm$1%) MeV beam were performed on a CentOS7 Linux distribution on the hpc Euler cluster of ETHZ using TOPAS version 3.7 and triplicates of $3.33 \times 10^5$ histories.

For the nanoscopic simulations, a simple vesicle model consisting of a 400 nm sized water sphere and randomly distributed 50 nm nanoparticles was centered in a water cube of 20 μm side length (X-rays) or 6 μm side length (protons). The dose inside the vesicle was scored as well as the dose around the vesicle within 100 nm water shells up to a distance of 1 μm. The beam size was 4 μm, ensuring an even irradiation field also around the vesicle. 50 nm sized nanoparticles were chosen, since for physical stage simulations smaller-sized nanoparticles only have a comparably small influence on the surrounding energy deposition[38], however, they increase computation time tremendously. The materials of the nanoparticles were defined by using the metal and oxygen or nitrogen mass content as given in Supplementary Table 1. For proton simulations it was important to only score and compare the dose in the nanoparticle-free space. For the microscopic dose-enhancement simulations, a simple cell geometry consisting of an outer water sphere with a diameter of 6 μm and an inner water sphere with a diameter of 3 μm was modeled representing the cytosol and the nucleus, respectively. This was close to the cytosol/nucleus ratio of 1.7 as used by Lin et al.[83] and Rudek et al.[38]. For X-ray simulations the cell was centered in a cubic water box of 20 μm side length, while for proton simulations it was centered in a smaller, 6 μm long cubic water box to reduce computation time. In the cell model all nanoparticles were placed in vesicles of 400 nm diameter at a pre-defined position and a volume fraction of 32.4%. Different amounts of such nanoparticle filled vesicles were then placed in the cytoplasm only, because metal oxides or Au nanoparticles enter cells predominantly by endocytotic pathways and are clustered within roughly 300–500 nm-sized vesicles within the cytoplasm, rarely entering the cell nucleus[31,84]. The particle beam (X-ray or proton) had the same diameter as the cell.

**Materials**. The following metal precursors and solvents were used: Titanium (IV) isopropoxide (Sigma-Aldrich, 97%), hafnium (IV) isopropoxide isopropanol adduct (Alfa Aesar, 99%), ammonium (meta)tungstate hydrate (Sigma-Aldrich, ≥85.0%), 2-ethylhexanoic acid (2-EHA) (Sigma-Aldrich, 99%), xylene (Sigma-Aldrich, ≥98.5%), ethanol absolute (VWR chemicals, ≥99.8%), diethylene glycol monobutyl ether (Sigma-Aldrich, ≥98.0%). 50 nm Au nanospheres (Citrate, Bio-Pure™) in USP purified water (1 mg/mL) were purchased from nanoComposix Inc (San Diego, USA). Hydrophilic fumed silicon dioxide nanoparticles (Aerosil®, A380) were obtained from Evonik Industries AG (Essen, Germany).

**Nanoparticle synthesis**. $TiO_2$, $HfO_2$, and $WO_3$ nanoparticles were synthesized using flame spray pyrolysis, as described in detail elsewhere[31,85,86]. The precursor to produce $TiO_2$ nanoparticles was made by stirring titanium isopropoxide and xylene at a metal concentration of 0.16 M for 30 min. Likewise, the $WO_3$ precursor solution was prepared by mixing ammonium (meta)tungstate hydrate with diethylene glycol monobutyl ether and ethanol (1:1) at a metal concentration of 0.2 M. In the case of $HfO_2$, hafnium isopropanol adduct was first dissolved in 2-EHA by stirring at 120 °C under reflux cooling for several hours at 0.96 M. After cooling

down, it was diluted 1:5 with xylene, reaching a final metal concentration of 0.16 M. The liquid precursor solutions were fed through a capillary with flow rates of 3 mL/min ($HfO_2$, $TiO_2$) or 5 mL/min ($WO_3$) and dispersed by $O_2$ (PanGas, purity > 99%) with a flow rate of 5 L/min into fine droplets. The pressure drop at the capillary was kept constant at 1.6 bar. A premixed ring-shaped $CH_4/O_2$ flame (1.5 L min$^{-1}$/3.2 L min$^{-1}$) ignited and stabilized the spray flame. Particles formed in the gas phase were collected on a glass fiber filter (Type GF6, Hahnemühle FineArt GmbH) with the aid of a vacuum pump (Busch Mink MM 1202 AV). The collected nanoparticle powder from the filter was subsequently sieved (mesh size = 250 μm) to remove any filter residues. Titanium nitride (TiN) nanoparticles were made from FSP $TiO_2$ nanoparticles following a previously established nitridation method[87]. Titania particles were nitrided on quartz wool in a U-shaped quartz reactor under pure ammonia flow (75 mL/min) and 700 °C heat treatment. The first heating rate up to 600 °C was 20 °C/min, followed by a second heating rate of 3 °C/min until the target temperature of 700 °C was reached and held for 30 h. After heat treatment the powder was cooled down to room temperature with a rate of 40 °C/min and soft oxidized (using 5% oxygen in argon) at room temperature.

**Nanoparticle characterization**. Transmission electron microscopy (TEM) imaging was performed on a JEOL 2200FS TEM operated at 200 kV. TEM samples were prepared by drop-casting dispersions of 100 μg/mL in ultrapure water or EtOH onto carbon-coated grids (200 mesh copper, EM Resolutions). X-ray diffraction patterns were obtained with a Bruker 2D Phaser with a step size of 0.01°. Rietveld refinement was done using Profex[88] (Version 4.2.5). XRD patterns as well as refinement results are plotted in Supplementary Fig. 12. The particles' specific surface area (SSA) was measured based on $N_2$ adsorption at 77 K using the Brunauer–Emmett–Teller (BET) (Tristar II Plus, Micrometrics) method. Prior to the measurement, all samples were degassed at 150 °C in $N_2$ for at least 45 min. The primary particle diameter, $d_{BET}$, was estimated using equation $d_{BET}[nm] = \frac{6000}{SSA[m^2 g^{-1}] \rho [g\, cm^{-3}]}$. Bulk material densities are given in Table 1. Hydrodynamic size measurements were acquired via dynamic light scattering (DLS) on a Zetasizer Nano ZS90 (Malvern Instruments Ltd., Worcestershire, UK) at a 90° scattering angle and a concentration of 0.1 mg/ml in ultrapure water. All nanoparticle suspensions were bath sonicated for at least 15 min prior to the measurement. To assess organic surface content, thermogravimetric analysis (TGA) was performed with a NETZSCH TG 209 F1 instrument (NETZSCH-Gerätebau GmbH, Selb, Germany) with a heating rate of 10 °C/min under nitrogen flow. The TGA as well as their first differential (DTG) can be found in Supplementary Fig. 13.

**Cell lines and culture conditions**. Human fibrosarcoma HT1080 cells (ATCC®CCL121TM) were cultured in minimum essential medium Eagle (MEM, Sigma-Aldrich or Gibco) supplemented with 10% fetal bovine serum (FBS, Sigma-Aldrich), 1% L-glutamine (Sigma-Aldrich), 1% penicillin–streptomycin (PS, Sigma-Aldrich), 1% non-essential amino acids (NEAA, Sigma-Aldrich), and 1 mM sodium-pyruvate at 37 °C under a humidified atmosphere containing 5% $CO_2$. Subculturing was conducted at 70–80% confluency by treatment with 0.5% Trypsin-EDTA (Sigma-Aldrich).

**Cell sample preparation for STEM imaging**. To image intracellular nanoparticles, $1.4 \times 10^5$ cells were seeded in T25 flasks in 8 mL medium and allowed to attach overnight. 700 μL of a 1 mg/mL nanoparticle suspension in ultrapure water were added. Following an incubation time of 24 h, the cells were washed twice with PBS, detached with 1 mL of Accutase (Sigma-Aldrich), washed with PBS and fixed overnight with 2.5% glutaraldehyde and 4% paraformaldehyde in PBS. The cell pellets were then washed with 0.1 M cacodylate buffer for 20 min and stained with 2% osmium tetroxide and 1.5% potassium ferricyanide for 1 h. Gradual dehydration via an ethanol gradient (50%, 70%, 80%, 90%, 100%, 100%, 100%) was followed by embedding into an epoxy resin (EPON 812 kit, Sigma-Aldrich). This was accomplished by mixing and adding resin and ethanol 1:2 (5 days), 1:1 (3 h), 1:0 (3 h), 1:0 (24 h) to the pellets. The resin blocks were then hardened in an oven at 60 °C for 48 h. Afterwards, thin sections of ~100 nm were cut from the blocks using an ultramicrotome (Scope M). The sections were imaged using a Hitachi S-4800 scanning electron microscope operated in transmission mode at 20 or 30 kV.

**Cell viability assessment and number of cell quantification**. The viability of cells was using the CellTiter-Glo (CTG) assay (CellTiter-Glo® Luminescent Cell Viability Assay, Promega, G7571) according to the manufacturer's specifications but adapted to our set-up. CTG buffer and substrate were thawed and mixed, and 300 μL of cell medium (from the experimental wells) was replaced by 200 μL CTG reagent. After incubation in the dark on a shaker for 20 min and equilibration for 30 min in the dark, luminescence (integration time 1 s) was measured with a microplate reader (Mithras LB 943 Multimode). An in-house made black titanium adapter was applied to eliminate crosstalk between the wells. This method was established by comparing the luminescence from 20,000 cells per well with and without adapter (Supplementary Fig. 14a). It was found that the adapter eliminated luminescence cross-talk, which led to an even distribution of the luminescence signal on the plate. A luminescence standard deviation of 5.0% with adapter compared to 13.9% without adapter was achieved. A cell standard curve with $N = 3$

independent biological experiments with duplicates was performed and with this we were able to convert the luminescence into number of cells (#cells) via: cells = $x = \frac{yb}{a-b}$, where $y$ is the luminescence signal and $a = 3.872 \times 10^6$ and $b = 3.431 \times 10^5$ are fitting parameters ($R^2 = 0.991$) (for more details see Supplementary Fig. 14b).

**Irradiation conditions.** A PMMA phantom consisting of two slabs of equal size ($4 \times 40 \times 40$ cm³) was prepared in-house and used for all X-ray experiments. A central recess was designed to fit a 48-well plate (TPP, Techno Plastic Products AG, Switzerland). Thus, photons traveled through ~3 cm PMMA phantom material before hitting the top of the 48-well plate (Supplementary Fig. 15a). For kV X-ray irradiation, a tube source (Seifert ISOVOLT 450, GE Sensing & Inspection Technologies GmbH, Germany) with a 7 mm beryllium filter window was positioned 50 cm above the bottom phantom slab and operated at 150 kV and 20 mA. The dose rate to the cell plate was ~1.5 Gy/min. A calibrated ionization chamber (N31003, PTW, Freiburg, Germany) was guided via an 8 mm inlet to the central recess and was connected to a UNIDOS dosimeter (PTW, Freiburg, Germany) to measure the dose rate within the phantom before cell irradiation, as well as during irradiation to ensure dose supply. MV X-ray irradiation was carried out at the hospital on a clinical linear accelerator (TrueBeam®, Varian, Paolo Alto, CA) at 6 MV with a field size of $20 \times 20$ cm² and a dose rate of 5.7 Gy/min. The distance between the beam focus and the upper slab of the phantom was 100 cm. Irradiation planning was done via the eclipseTM treatment planning system with the Acuros® algorithm (Varian, Paolo Alto, CA) and CT imaging (SOMATOM Definition, Siemens, Erlangen, Germany) of the plate within the phantom. Measurements with EBT3 dosimetry films confirmed the accuracy of the dose calculations and that homogeneity was high with a standard deviation of 0.82%. A 4 Gy irradiated films was cross-validated with one from a kV-irradiated EBT3 film. Proton irradiations (Supplementary Fig. 15b) were carried out at the Gantry 2 (PSI, Paul Scherrer Institute, Villigen, Switzerland) with a 100 MeV ($\Delta p/p = 1\%$) proton beam. The plateau region of the Bragg peak curve in shoot-through mode employing the spot-scanning technique was used to deliver a uniform dose field. The irradiation field size was $10 \times 15.2$ cm² with a spot distance of 0.4 cm in $x$- and $y$-direction. An Advanced Markus ionization chamber (PTW, Freiburg, Germany) was used to verify dose delivery better than 1% and was confirmed by additional EBT3 film measurements.

**Acellular ROS assay and chemical dose-enhancement quantification.** The DCF assay was adapted from Zhao and Riediker (2014)[89]. 2′,7′-dichlorodihydrofluorecein diacetate (H2DCF−DA) powder (Sigma-Aldrich) was dissolved in dimethyl sulfoxide (DMSO, Sigma-Aldrich) to reach a concentration of 5 mM. Aliquots of this were stored at −20 °C for further use. The thawed H2DCF-DA stock solution was mixed with NaOH (10 mM) 1:4 and incubated in the dark for 30 min. Thereafter, it was diluted with Tris–HCl buffer (0.1 M, pH = 7.4) to 8 µM. The final H2DCF working solution was kept on ice in the dark at all times. All nanoparticles were weighted, dispersed and bath sonicated in Tris–HCl buffer reaching 10× stock concentrations. Prior to irradiation, 0.1 mL Tris–HCl buffer, 0.1 mL nanoparticle-suspension and 0.8 ml H2DCF working solution were added to the wells of two transparent 24-well plates identically. Final nanoparticle concentrations were of 36.8, 368, and 1840 cm²/mL. One plate served as 0 Gy reference plate, the other plate was irradiated in the phantom with 12 Gy. Following irradiation, the liquid from the wells was centrifuged in microtubes and triplicates of the supernatant were transferred to transparent 96-well plates. The fluorescence signal was measured using a microplate reader (485 nm excitation, 535 nm emission, Tecan infinite 200Pro or Mithras LB 943 Multimode). The average 0 Gy fluorescence values, $FI_{0Gy}$, of each particle or blank were subtracted from the 12 Gy averages, $FI_{12Gy}$. The dose-enhancement factors ($DEF_{ROS}$) were then calculated using the fluorescence intensity (FI) as follows:

$$DCF_{ROS} = \frac{FI_{12Gy\ with\ NP} - FI_{0Gy\ with\ NP}}{FI_{12Gy\ without\ NP} - FI_{0Gy\ without\ NP}} \quad (3)$$

**Cell irradiation and in vitro dose-enhancement quantification.** 2000 HT1080 cells in 300 µL cell medium were seeded in 48 well plates and led to adhere for 24 h. After this, cells were incubated with 200 µL of different nanoparticle or control solutions to reaching final nanoparticle concentrations of 0–320 µg/mL. To achieve different nanoparticle concentrations for cell incubation, solutions were prepared from a fresh and 30 min bath sonicated 3.2 mg/mL nanoparticle-milliQ stock solution which was added in different volumes to a fixed amounts of cell medium. MilliQ was used to adjust the solutions to achieve the right concentration and a fixed amount (10%) of water in every solution. Nanoparticle free solutions consisted also of cell medium and 10% milliQ water. After 24 h cells were washed twice using 250 µL PBS. 500 µL cell medium was added thereafter. For the transport to and from the different irradiation facilities cells were kept in a cooled box. Irradiation was performed at room temperature. After the irradiation procedure, cells were incubated at standard conditions and medium was changed 2 days after. On the 5th day after irradiation, cell viability was assessed using the CellTiter-Glo® assay. One biological repeat per nanoparticle concentration consisted of triplicates ($n = 3$), and for control cells of sextuplicates ($n = 6$). To quantify in vitro dose

enhancement, first the luminescence from the viability assessment was translated into number of cells (#cells) as stated above. From that the surviving fraction, SF, of cells following the irradiation dose responses of 0, 2, 4, 6, and 8 Gy, was calculated as

$$SF(xGy) = \frac{cells_{xGy,with\ or\ without\ NPs}}{cells_{0Gy,with\ or\ without\ NPs}} \quad (4)$$

The data was then fitted using MATLAB (MathWorks Inc., MA, US) to the linear quadratic model

$$SF(D) = e^{-(\alpha D + \beta D^2)} \quad (5)$$

where $D$ is the irradiation dose in Gy and $\alpha$ and $\beta$ are fitting parameters. The dose-modifying factor was extracted by finding the lethal dose leading to 50% survival ($LD_{50\%}$) with any particle type and comparing it to the one without nanoparticles:

$$DMR_{50\%} = \frac{LD_{50\%,without\ NPs}}{LD_{50\%,with\ NPs}} \quad (6)$$

Sham-irradiation toxicity of nanoparticles was defined by normalizing the number of cells treated with nanoparticles to the number of control cells treated with no nanoparticles and irradiated with 0 Gy:

$$Normalized\ viability = \frac{\#cells_{0Gy,\ with\ NPs}}{\#cells_{0Gy,\ without\ NPs}} \quad (7)$$

To be included in the $DMR_{50\%}$ analysis, a sham irradiation cell viability cut-off of 60% (lowest limit) was defined for all nanoparticles. A sigmoidal curve with fitting parameters $a$ and $b$ in the form of

$$Normalized\ viability = 1 - \frac{1}{1 + ae^{-bx}} \quad (8)$$

was fitted to the cell-viability data to extract nanoparticle LD50 doses ($x$).

**Cell digestion and nanoparticle uptake quantification using ICP-MS.** To analyze nanoparticle uptake into cells, cells were seeded and treated in duplicates in the same way as in a typical irradiation experiment. After 24 h nanoparticle treatment, cells were washed twice with 250 µL PBS. Thereafter, instead of adding cell medium, cells were trypsinized (80 µL). Trypsinration was stopped using 220 µL medium and cells were transferred into Eppendorf tubes and stored at −20 °C until further analysis. To acquire the cell number, three experimental wells with control cells were pooled, centrifuged at $200 \times g$ for 5 min and counted using a hematocytometer. This was repeated twice. Prior to ICP-MS analysis, cells were digested in 1 mL HNO₃, 3 mL HCl and 0.5 mL HF using a 1 h bath sonication treatment. After digestion, samples were filled up to 50 mL using MilliQ water. For ICP-MS elemental standard curves, ionic metal standards (Au, Hf, W, and Ti) were prepared in the same matrix as the samples in concentrations from 0.01 to 10 ppb. A 1 ppb quality control (IV4 or IV100, Inorganic Ventures, Christiansburg, VI, USA) served as reference and ensured correctness of the standard curve. Measurements were performed on an Agilent 7900 ICP-MS (Agilent Technologies, Santa Clara, USA). Non-spectral interference effects were corrected using 100 ppb Sc, Ge, In, and Lu mixed on-line with the sample via a t-piece at a ratio of 1:10. All elements were measured in Helium mode. Elemental masses were then normalized to cell number and further translated into nanoparticle volume fraction using the atomic mass fraction as well as the nanoparticle bulk density as specified in Table S1 of the Supplementary information. The nanoparticle volume fraction was then corrected using the sham-irradiation viability fit from Supplementary Fig. 1. The volume of a cells was defined as 2800 µm³ using the diameter of 17.5 µm as given by a coulter counter. The relationship of metal mass per cell ($f(x)$) and nominal nanoparticle concentration ($x$) was described using a non-linear fit with fitting parameters A, B, and $p$ in the form of

$$f(x) = \frac{Ax^p}{B^p + x^p} \quad (9)$$

**Hydroxyl radical scavenging using DMSO.** To study the effect of hydroxyl radial quenching during irradiation with or without nanoparticles, DMSO in cell medium at different concentrations (0, 0.11, 0.334, 0.667, 1 M) was added (in quadruplicates, $n = 4$) prior to 0 or 6 Gy irradiation and replaced by cell medium after irradiation, before putting the cells back into the incubator. Similar to the other irradiation experiments, 2000 cells were initially seeded into the inner wells of 48-well plates, left to adhere for 24 h, and medium mixes with or without pre-sonicated nanoparticles were added to the wells. A no-nanoparticle control was added on every plate. The outer wells of the plates were filled with 500 µL sterile PBS to simulate the same humidity in every well. The final water content was fixed at 10% in all experimental wells. After 24 h cells were washed as usual, before DMSO-treated medium was added and 150 kVp X-ray irradiation took place. Cell viability was analyzed 5 days after irradiation, as usual. The degree of DMSO protection (DoP) was defined as the difference in the nanoparticle-related

enhancement effect with versus without DMSO and calculated as following:

$$DoP = 1 - \left( \frac{SF_{6Gy, \, DMSO, \, no \, NP}}{SF_{6Gy, \, DMSO, \, NP}} - 1 \right) \Big/ \left( \frac{SF_{6Gy, \, no \, DMSO, \, no \, NP}}{SF_{6Gy, \, no \, DMSO, \, NP}} - 1 \right) \quad (10)$$

using the survival fraction, SF, of cells at 6 Gy as defined above.

**Reporting summary**. Further information on research design is available in the Nature Research Reporting Summary linked to this article.

## Data availability

The authors declare that the data supporting the findings of this study are available within the manuscript and its supplementary information files. The raw data from the Monte Carlo simulations are provided with this paper. Processable raw data is available from the corresponding author upon request. Source data are provided with this paper.

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

## Acknowledgements

This study was funded in parts by the Swiss National Science Foundation (Eccellenza grant no. 181290, I.K.H.), the Swiss Cancer League (KFS-4868-08-2019, I.K.H.) and an ETH Grant (ETH-07 21-2, I.K.H.). We thank Prof. S.E. Pratsinis for access to PTL infrastructure, Ralf Kägi and Andreas Voegelin for access and Brian Sinnet for assistance at the HF digestion facilities of the Swiss Federal Institute of Aquatic Science and Technology (EAWAG), and Pascal M. Gschwend for the supply of TiN nanoparticles. Parts of the schematics in Supplementary Fig. 15a, b were created with BioRender.com.

## Author contributions

L.R.H.G. contributed to the study design, performed experiments and simulations, analyzed data and drafted the manuscript. A.G. developed and performed elemental analysis protocols. F.H.L.S. helped with ROS assay design and nanoparticle synthesis. H.D. developed the ROS assay protocol. H.D. and M.E.G. performed ROS experiments and synthesized nanoparticles. H.S., S.P. and D.M. helped with irradiation experiments and verified the dose supply during photon or proton irradiation. D.C.W. and L.P. provided input on the experiments at the clinical irradiation facilities. I.K.H. conceived and supervised the study and edited the manuscript. All authors contributed to the manuscript writing and have given approval to the final version of the manuscript.

## Competing interests

The authors declare no competing interests.
