## [Peer Review File · Nature Communications]

REVIEWER COMMENTS

Reviewer #1 (Remarks to the Author):

The effects of ionising radiation on biological structures are governed by physical, chemical and biological phenomena and the exact contributions of these different effects are complex to determine. The mechanistic understanding is particularly hampered by the lack of fundamental and comparative studies, which prevents the rational design of nanoparticle-based radio-enhancers. Although a large number of studies have been published concerning nanoparticles and X-ray radiotherapy, studies comparing a large number of nanoparticles in the same cell line and with different radiation qualities are few, especially with protons, which makes these results particularly interesting.

This study proposes to provide elements of mechanistic understanding by studying 6 nanoparticles of different nature on a single cell line and irradiating them with different types of radiation: X-ray 150 kVp, 6 MV Xrays and proton beams. The catalytic processes are also studied and related to the physical effects of the dose increase. Finally, a simulation study also provides elements of comparison between the elements with respect to the physical dose enhancement.

This is an interesting study that brings together a very large number of experimental results. The experiments are well documented and the results well presented. Overall, the results are well discussed and the conclusions are relevant to the results presented.

Most of the time, nanoparticles are coated to make the nanoparticles usable in in vivo studies and to improve their tolerance. Could you discuss the potential influence of the coating on the catalytic and physical enhancement effects?

In vivo, the overall efficacy of nanoparticles under irradiation will depend greatly on the concentrations of nanoparticles, and their extra and/or intra cellular localization. This point should be further discussed.

Specific comments:

Page 9: Define the DEF

Page 10 : There is an error in the definition of the macroscopic DEF:

The “mass absorption energy coefficient” has to be taken into account and not the “mass energy attenuation coefficient”, as stated.

Page 16 : typo error « Ratio at 50% cell xxx » and not “Ration at 50% cell xxx “

Page: 31 specify also the x-ray tube filtration.

Specify on which side of the PMMA phantom the beam was directed.

The graphs showing a comparison of the results against the different elements should be presented in the same format (the elements in the same order). Figures 4 F, 6 F, S5B, S9D

Supplementary data : Table S1 : add the Z of the elements.

Reviewer #2 (Remarks to the Author):

The clarification of the radiation dose enhancement processes due to the nanoparticle presence in the tumor radiation therapy is a topic research field at moment. In this paper the authors investigated all aspects suspected of taking part in this process. It is very interesting and highlights new information. However, I have same question that I think it would be better to clarify.

Pag. 4: the sentence “While photons deposit energy continuously” sounds very strange to me. Maybe the authors would say that the dose deposition of photons in depth is continuous and goes beyond the tumor. Instead the dose deposition of protons have a peak at the end of the proton range related to the energy....

Pag.7: As indicated in different research work on the same topic the results are related to the cellular line used. Please add comments on this aspect.

Page 10: Please specify fvol

Pag. 10. The maximum enhancement is found when the nanoparticles occupy 33.4 of the vesicle volume. How do you relate the amount of volume occupied in the vesicle with the amount of material administered to the culture?

In standard experiment, how do you know with what percentage the nanoparticles are distributed in the cytoplasm and in the nucleus? Does this depend on the cell line? Does it depend on the size of the nano? There are works that say it never enters the nucleus. Please comment on this aspect.

Pag. 27: without a scratch of the irradiation geometry is very hard to follow the description. Please could you add a new figure with the irradiation geometries?

Pag. 27: “nanoparticle vesicles were placed in the cytoplasm only” ..., I understood that the simulation were realized also with the nanoparticle presence in the nucleus. Please clarify this aspect.

Reviewer #3 (Remarks to the Author):

Radioenhancement by nano-particles is dicussed for decades as a promissing procedure to locally increase radiation damage in tumor cells while reducing the generfal radiation load on healthy cells. However up to now experimental data are often contradicting. The manuscript of Gerken et al. is addressing this situation and presenting systematic simulation and experimental studies. They systematically investigate enhancement effects of metal oxide nanoparticles and nanogold under different radiation conditions especially for MV photons and protons.

The article is well written and the results are supported by additional data in the supplement. I recommend publication after minor revision.

1.) What are the noteworthy results?

The results for therapeutic energies of 6MV photons and 100 MeV protons

2.) Will the work be of significance to the field and related fields?

Yes, the work is highly significant and gives recommendations for nanoparticle design

3.) How does it compare to the established literature? If the work is not original, please provide relevant references.

It is original work.

However and this is my concern for revision: Recently new results were published* describing effects and data of nanogold dose enhancement and mechanisms behind. These publications should be considered and appropriately included in the discussion of the results.

*Dobešová et al., Incorporation of Low Concentrations of Gold Nanoparticles: Complex Effects on Radiation Response and Fate of Cancer Cells. *Pharmaceutics* 2022, 14, 166. <https://doi.org/10.3390/pharmaceutics14010166> and Pagáčová et al., Challenges and Contradictions of Metal Nano-Particle Applications for Radio-Sensitivity Enhancement in Cancer Therapy. *Int. J. Mol. Sci.* 2019, 20, 588; doi:10.3390/ijms20030588

4.) Does the work support the conclusions and claims, or is additional evidence needed?

Yes it does

5.) Are there any flaws in the data analysis, interpretation and conclusions? Do these prohibit publication or require revision?

No, well performed analysis.

6.) Is the methodology sound? Does the work meet the expected standards in your field?

up-to-date standards

7.) Is there enough detail provided in the methods for the work to be reproduced?

Yes

Title: Catalytic activity imperative for nanoparticle dose enhancement in photon and proton therapy

Author(s): Lukas Gerken *et al.*

Letter of Reply

We thank the reviewers for their careful and positive assessment of our manuscript. Please find a detailed reply to the respective comments and concerns below. Changes to the manuscript text are highlighted.

Reviewer #1:

The effects of ionising radiation on biological structures are governed by physical, chemical and biological phenomena and the exact contributions of these different effects are complex to determine. The mechanistic understanding is particularly hampered by the lack of fundamental and comparative studies, which prevents the rational design of nanoparticle-based radio-enhancers. Although a large number of studies have been published concerning nanoparticles and X-ray radiotherapy, studies comparing a large number of nanoparticles in the same cell line and with different radiation qualities are few, especially with protons, which makes these results particularly interesting.

This study proposes to provide elements of mechanistic understanding by studying 6 nanoparticles of different nature on a single cell line and irradiating them with different types of radiation: X-ray 150 kVp, 6 MV Xrays and proton beams. The catalytic processes are also studied and related to the physical effects of the dose increase. Finally, a simulation study also provides elements of comparison between the elements with respect to the physical dose enhancement.

This is an interesting study that brings together a very large number of experimental results. The experiments are well documented and the results well presented. Overall, the results are well discussed and the conclusions are relevant to the results presented.

1. Most of the time, nanoparticles are coated to make the nanoparticles usable in in vivo studies and to improve their tolerance. Could you discuss the potential influence of the coating on the catalytic and physical enhancement effects?

We thank the reviewer for raising this point and we agree that this is a very important aspect, which we plan to investigate in our future research. We have added a short paragraph discussing potential implications on page 27:

Potential effects of nanomaterial surface functionalization on dose-enhancement should be investigated carefully, including potential ROS quenching by antioxidant molecules (such as dopamine), as well as potentially synergistic effects leading to augmented ROS generation, e.g. by porphyrins.

2. In vivo, the overall efficacy of nanoparticles under irradiation will depend greatly on the concentrations of nanoparticles, and their extra and/or intra cellular localization. This point should be further discussed.

We have added a short paragraph discussing effects of biodistribution on page 27:

Eventually, the effective nanoparticle concentration reached in the cancer cells will govern dose enhancement. While experimental research in mouse models on intravenously injected nanoparticles has indicated that only a small fraction of the injected nanoparticles may accumulate inside cancer cells, intratumoral administration of HfO₂ nanoparticles (NBTXR3) appears to partially overcome the delivery problem and shows convincing therapeutic effectiveness in preclinical and clinical settings. This effectiveness may, however, be even further improved by optimizing radioenhancer designs, in parts based on the insights provided by this work.

3. Page 9: Define the DEF

We have added the definition on page number 9: "We extracted physical dose enhancement factors (DEFs) by building the ratios of the dose scored to the cytosol, nucleus, vesicle or water shells, respectively, in the presence of the nanoparticles to that with no nanoparticles (water)."

4. Page 10: There is an error in the definition of the macroscopic DEF: The "mass absorption energy coefficient" has to be taken into account and not the "mass energy attenuation coefficient", as stated.

We have corrected the definition and replaced the term "mass energy attenuation coefficient" by "mass energy-absorption coefficient" according to the terms specified by the National Institute of Standards and Technology.

The following correction can be found on page 11: $DEF_{macroscopic} = 1 + f_z \cdot \left(\frac{\mu_{en}(E)}{\rho}\right)_Z / \left(\frac{\mu_{en}(E)}{\rho}\right)_{H_2O}$, where f_z is the atomic number (Z) mass fraction in the system and $\mu_{en}(E)/\rho$ the mass energy absorption coefficient at a monoenergetic photon energy, E."

5. Page 16: typo error "Ratio at 50% cell xxx » and not "Ration at 50% cell xxx"

We have corrected this on page 16: "... by calculating the Dose Modifying Ratio at 50% cell survival ..."

6. Page 31: specify also the x-ray tube filtration. Specify on which side of the PMMA phantom the beam was directed.

We have specified this further on page 33: "Thus, photons travelled through approximately 3 cm PMMA phantom material before hitting the top of the 48-well plate. For kV X-ray irradiation, a tube source ... with a 7 mm beryllium filter window was positioned..." We have also added a schematic figure illustrating the irradiation set up in the supplementary information in Figure S15.

7. The graphs showing a comparison of the results against the different elements should be presented in the same format (the elements in the same order). Figures 4F, 6F, S5B, S9D

We have harmonized the presentation of these figures.

8. Supplementary data: Table S1: add the Z of the elements.

We have added the atomic numbers (Z) of the elements in Table S1 of the supplementary information.

Reviewer #2:

The clarification of the radiation dose enhancement processes due to the nanoparticle presence in the tumor radiation therapy is a topic research field at moment. In this paper the authors investigated all aspects suspected of taking part in this process. It is very interesting and highlights new information. However, I have same question that I think it would be better to clarify.

1. Pag. 4: the sentence "While photons deposit energy continuously" sounds very strange to me. Maybe the authors would say that the dose deposition of photons in depth is continuous and goes beyond the tumor. Instead the dose deposition of protons have a peak at the end of the proton range related to the energy.

We have adapted this statement on page 4: "While the dose deposition of photons in depth is continuous and goes beyond the tumor resulting in an "exit dose", protons lose the majority of their energy in the range of the Bragg peak, after which they are stopped completely."

2. Pag.7: As indicated in different research work on the same topic the results are related to the cellular line used. Please add comments on this aspect.

We have added a paragraph to page 26:

While nanoparticle uptake, cellular toxicity, and radio-enhancement are cell-line dependent, previous work on different cancer cell lines has indicated that the relative trends in radio-enhancer effectiveness hold true, albeit with slightly different absolute values.³¹

3. Page 10: Please specify f_{vol}

We have specified f_{vol} on page 11: "Mass and volume fraction (f_{vol}) are related via a constant density ratio of the materials."

4. Pag. 10. The maximum enhancement is found when the nanoparticles occupy 32.4 of the vesicle volume. How do you relate the amount of volume occupied in the vesicle with the amount of material administered to the culture? In standard experiment, how do you know with what percentage the nanoparticles are distributed in the cytoplasm and in the nucleus? Does this depend on the cell line? Does it depend on the size of the nano? There are works that say it never enters the nucleus. Please comment on this aspect.

The intra-vesicular enhancement increases with increasing nanoparticle volume fraction. By packing a vesicle randomly with spheres, a maximum packing fraction was reached at 32.4 %. Such volume fractions are also observed experimentally, for example by studying nanoparticle uptake using liquid scanning transmission electron microscopy.¹ We have added a paragraph to page 10:

The dose enhancement factors within a nanoparticle-filled vesicle reached values of DEF = 30-40 for Au nanoparticles and DEF = 10-20 for HfO₂ and WO₃ nanoparticles at the highest reached nanoparticle content of 32.4 vol% (volume percent) in the vesicle (Figure 3A). This packing fraction is also reasonable for biological scenarios. For instance, nanoparticle volume fractions of $35 \pm 16\%$ per vesicle have been reported in cells for 30-nm sized Au nanoparticles,³⁶ and exposure conditions similar to the ones used in our study.

No particles could be detected in the nucleus and all particles were distributed in the cytosol. We have added the following statement to page 8:

Few hundred nanometer up to micrometer sized nanoparticle agglomerates were distributed within the cell cytoplasm (in vesicles or endosomes). In the > 100 cells analyzed per nanoparticle type, no evidence for nanoparticle uptake into the nucleus was found, even though uptake overall, and nanoparticle accumulation in the nucleus, might be particle and cell type dependent.^{2,3}

5. Pag. 27: without a scratch of the irradiation geometry is very hard to follow the description. Please could you add a new figure with the irradiation geometries?

We have added a schematic to the Supplementary Information as Figure S15.

6. Pag. 27: "nanoparticle vesicles were placed in the cytoplasm only" ..., I understood that the simulation were realized also with the nanoparticle presence in the nucleus. Please clarify this aspect.

As the reviewer has pointed out correctly in comment number 4, nanoparticles rarely enter the cell nucleus. Therefore, we have performed our simulations under the assumption, that all nanoparticles are distributed in agglomerates within the cytoplasm only. This assumption is reflected in our TEM observations in this and earlier work,^{4,5,6} Nevertheless, we have scored the dose deposited within the cell nucleus, while particles were only present in the cytoplasm. To clarify this further, we have modified the paragraphs on page 9:" The geometries were built to match cellular uptake scenarios as closely as possible, with ~ 400s nm nanoparticle agglomerates distributed only within the cytosol (see also Figure 2).³¹ As nanoparticle uptake into the nucleus was not observed experimentally, it was considered negligible also for the simulations." and on page 29: "Different amounts of such nanoparticle filled vesicles were then placed in the cytoplasm only, because metal oxides or gold nanoparticles enter cells predominantly by endocytotic pathways and are clustered within roughly 300 – 500 nm sized vesicles within the cytoplasm, rarely entering the cell nucleus."

Reviewer #3:

Radioenhancement by nano-particles is discussed for decades as a promising procedure to locally increase radiation damage in tumor cells while reducing the general radiation load on healthy cells. However up to now experimental data are often contradicting. The manuscript of Gerken et al. is addressing

this situation and presenting systematic simulation and experimental studies. They systematically investigate enhancement effects of metal oxide nanoparticles and nanogold under different radiation conditions especially for MV photons and protons.

The article is well written and the results are supported by additional data in the supplement. I recommend publication after minor revision.

1. What are the noteworthy results? The results for therapeutic energies of 6MV photons and 100 MeV protons.

2. Will the work be of significance to the field and related fields? Yes, the work is highly significant and gives recommendations for nanoparticle design.

3. How does it compare to the established literature? If the work is not original, please provide relevant references. It is original work. However and this is my concern for revision: Recently new results were published* describing effects and data of nanogold dose enhancement and mechanisms behind. These publications should be considered and appropriately included in the discussion of the results.

*Dobešová et al., Incorporation of Low Concentrations of Gold Nanoparticles: Complex Effects on Radiation Response and Fate of Cancer Cells. *Pharmaceutics* 2022, 14, 166. <https://doi.org/10.3390/pharmaceutics14010166> and

Pagáčová et al., Challenges and Contradictions of Metal Nano-Particle Applications for Radio-Sensitivity Enhancement in Cancer Therapy. *Int. J. Mol. Sci.* 2019, 20, 588; doi:10.3390/ijms20030588

We have added a paragraph discussing these studies on page 23:

Additionally, •OH radical formation in the cytosol might then be a combination of physical and surface catalytic processes for these high-Z materials at kV X-ray energies. Interestingly, cytoplasmic processes leading to the disruption of organelles, such as mitochondria or lysosomes, may play a major role in nanoparticle mediated radioenhancement.^{7,8} Most recently, it was also shown that even very low concentration of 10 nm gold nanoparticles can have an effect on cell cycle phase, the proportion of radiosensitive G2 cells as well as the DSB repair kinetics.⁹ Thus, the nanoparticle mediated radiation response is complex with sensitization of cancer cells as well as dose enhancement both contributing to the overall response.

4. Does the work support the conclusions and claims, or is additional evidence needed? Yes it does.

5. Are there any flaws in the data analysis, interpretation and conclusions? Do these prohibit publication or require revision? No, well performed analysis.

6. Is the methodology sound? Does the work meet the expected standards in your field? up-to-date standards

7. Is there enough detail provided in the methods for the work to be reproduced? Yes

REFERENCES (Letter of Reply):

1. Peckys, D. B. & de Jonge, N. Visualizing Gold Nanoparticle Uptake in Live Cells with Liquid Scanning Transmission Electron Microscopy. *Nano Lett.* **11**, 1733–1738 (2011).
2. Huo, S. *et al.* Ultrasmall Gold Nanoparticles as Carriers for Nucleus-Based Gene Therapy Due to Size-Dependent Nuclear Entry. *ACS Nano* **8**, 5852–5862 (2014).
3. Coulter, J. Cell type-dependent uptake, localization, and cytotoxicity of 1.9 nm gold nanoparticles. *IJN* 2673 (2012) doi:10.2147/IJN.S31751.
4. Gerken, L. R. H. *et al.* Scalable Synthesis of Ultrasmall Metal Oxide Radio-Enhancers Outperforming Gold. *Chem. Mater.* **33**, 3098–3112 (2021).
5. Peckys, D. B. & Jonge, N. de. Gold Nanoparticle Uptake in Whole Cells in Liquid Examined by Environmental Scanning Electron Microscopy. *Microscopy and Microanalysis* **20**, 189–197 (2014).
6. Chithrani, D. B. *et al.* Gold Nanoparticles as Radiation Sensitizers in Cancer Therapy. *rare* **173**, 719–728 (2010).
7. Pagáčová, E. *et al.* Challenges and Contradictions of Metal Nano-Particle Applications for Radio-Sensitivity Enhancement in Cancer Therapy. *Int J Mol Sci* **20**, 588 (2019).
8. Ghita, M. *et al.* A mechanistic study of gold nanoparticle radiosensitisation using targeted microbeam irradiation. *Sci Rep* **7**, 44752 (2017).
9. Dobešová, L. *et al.* Incorporation of Low Concentrations of Gold Nanoparticles: Complex Effects on Radiation Response and Fate of Cancer Cells. *Pharmaceutics* **14**, 166 (2022).

REVIEWERS' COMMENTS

Reviewer #1 (Remarks to the Author):

The authors have satisfactorily taken into account my remarks and I am now in favor of the publication of this manuscript.

Reviewer #2 (Remarks to the Author):

The authors responded to all my observations satisfactorily